# Diabetes and Its Cardiovascular Complications: Potential Role of the Acetyltransferase p300

**DOI:** 10.3390/cells12030431

**Published:** 2023-01-28

**Authors:** Nadia Di Pietrantonio, Pamela Di Tomo, Domitilla Mandatori, Gloria Formoso, Assunta Pandolfi

**Affiliations:** 1Department of Medical, Oral and Biotechnological Sciences, Center for Advanced Studies and Technology-CAST, University G. D’Annunzio of Chieti-Pescara, 66100 Chieti, Italy; 2Department of Medicine and Aging Sciences, Center for Advanced Studies and Technology-CAST, University G. D’Annunzio of Chieti-Pescara, 66100 Chieti, Italy

**Keywords:** diabetes, cardiovascular disease, endothelial dysfunction, senescence, p300, epigenetics, inflammation, oxidative stress

## Abstract

Diabetes has been shown to accelerate vascular senescence, which is associated with chronic inflammation and oxidative stress, both implicated in the development of endothelial dysfunction. This condition represents the initial alteration linking diabetes to related cardiovascular (CV) complications. Recently, it has been hypothesised that the acetyltransferase, p300, may contribute to establishing an early vascular senescent phenotype, playing a relevant role in diabetes-associated inflammation and oxidative stress, which drive endothelial dysfunction. Specifically, p300 can modulate vascular inflammation through epigenetic mechanisms and transcription factors acetylation. Indeed, it regulates the inflammatory pathway by interacting with nuclear factor kappa-light-chain-enhancer of activated B cells p65 subunit (NF-κB p65) or by inducing its acetylation, suggesting a crucial role of p300 as a bridge between NF-κB p65 and the transcriptional machinery. Additionally, p300-mediated epigenetic modifications could be upstream of the activation of inflammatory cytokines, and they may induce oxidative stress by affecting the production of reactive oxygen species (ROS). Because several in vitro and in vivo studies shed light on the potential use of acetyltransferase inhibitors, a better understanding of the mechanisms underlying the role of p300 in diabetic vascular dysfunction could help in finding new strategies for the clinical management of CV diseases related to diabetes.

## 1. Cardiovascular Disease and Diabetes

Cardiovascular diseases (CVDs) are the main causes of death and disability among patients with diabetes. This was confirmed by a meta-analysis of 102 prospective studies, showing that diabetes confers a two-fold excess risk in adverse vascular outcomes, independently of other risk factors. CV complications associated with diabetes can be classified as microvascular, such as nephropathy and retinopathy, which are major causes of kidney disease and blindness, respectively, or macrovascular complications, including ischaemic heart disease, heart failure, stroke, coronary artery disease and peripheral artery disease [1,2,3]. Specifically, macrovascular damage in diabetes is characterised by atherosclerotic disease affecting coronary, cerebral and peripheral arteries [4].

Diabetes-associated endothelial dysfunction is one of the most critical initiating triggers of vascular complications and, therefore, an important predictor of the onset of CVD.

This is a very complex process orchestrated by several factors, such as hyperglycaemia, dyslipidaemia, free fatty acids and insulin resistance, which impair the normal functionality of the endothelium by promoting inflammation, oxidative stress, thrombosis, arterial stiffness and the impaired regulation of arterial tone and flow [4]. All these factors contribute to generating an imbalance between vasodilating and vasoconstricting factors, favouring proinflammatory and prothrombotic effects that promote atherosclerosis. In detail, hyperglycaemia can induce a series of cellular events that increase the production of ROS, such as superoxide anion, which induces oxidative stress and, in turn, leads to a reduced NO bioavailability [5].

In addition to impaired vasodilation, diabetes is also associated with increased circulating levels of endothelium-derived adhesion molecules, plasminogen activator inhibitor-1 and high triglyceride levels, which denote a proinflammatory endothelial phenotype. Inflammation is also driven by the increase in advanced glycation end products (AGEs), which result from the overproduction of mitochondrial superoxide anion. Diabetes is also characterised by high circulating levels of free fatty acids, which may compromise endothelial function through the activation of protein kinase C and the dysregulation of insulin pathway transduction.

All these processes exacerbate vascular inflammation and are characterised by the increased expression of several cytokines, such as Vascular cell adhesion molecule 1 (VCAM-1), E-selectin, Intercellular adhesion molecule-1 (ICAM-1), Interleukin-6 (IL-6), IL-1 and IL-8 and Monocyte chemoattractant protein-1 (MCP-1), facilitating the adhesion of monocytes, neutrophils and macrophages to the endothelium and, consequently, the formation of atherosclerotic plaque [6,7,8,9,10,11,12,13,14].

Although diabetes management has improved, the prevention and control of cardiovascular complications represent the main target of diabetes therapy [15], and the mechanisms by which hyperglycaemia contributes to CV system alteration need further elucidation.

Diabetic patients present dysfunction of multiple organ systems similar to that observed in chronological ageing, suggesting that this metabolic disease might induce an early ageing state associated with early vascular senescence, implicated in the development of endothelial dysfunction [16,17,18,19,20,21,22,23]. Therefore, a deep understanding of the molecular mechanisms underlying diabetic premature vascular senescence could lead to identifying new potential targets in order to control the progression of vascular damage in diabetes.

## 2. Cardiovascular Senescence and Diabetes 

Cellular senescence is an irreversible arrest of the cell cycle, which has the purpose of repairing or eliminating damaged cells by restoring tissue homeostasis [24,25,26]. However, senescent cells can significantly increase, leading to biological ageing and the development of diseases. The process of senescence is triggered by various intrinsic and extrinsic stimuli, including genomic instability, epigenetic alterations, telomere modifications, mitochondrial dysfunction, stem cell exhaustion, mitogenic signals, oncogenic activation, radiation, oxidative stress, inflammation, tissue damage signals and nutrient deprivation [27]. All these stimuli lead to the upregulation of p53/p21, proteins with a central role in DNA repair and cell cycle regulation, p16/retinoblastoma protein (Rb) pathway activation and markers associated with activation of the DNA damage response (DDR), such as p38 mitogen-activated protein kinase (MAPK) and phosphorylated histone 2AX (cH2AX) [28,29].

Age increases the susceptibility to a wide variety of diseases, which can be considered as degenerative pathologies because they lead to the tissues’ normal functionality loss. Examples include neurodegenerative diseases, such as Alzheimer’s disease, Parkinson’s disease, Huntington’s disease, cardiovascular disease, musculoskeletal decrements and cancer [30,31]. Furthermore, evidence has shown that the senescence-associated secretory phenotype (SASP) can promote malignant phenotypes in culture and tumour growth in vivo [32]. Moreover, the accumulation of senescent cells has been also associated with the accelerated degeneration of bones and joints and skeletal muscle frailty [33].

Senescence has also been shown to be key in the pathophysiology of CVDs, such as atherosclerosis, myocardial infarction and cardiac fibrosis. Indeed, it was found that cardiac cells exhibit a SASP, which induces the release of soluble signalling factors, proteases and extracellular matrix (ECM) components in the surrounding environment that contribute to cardiovascular damage initiation and progression [34,35].

Numerous studies have shown that cardiovascular ageing is largely characterised by cardiomyocyte hypertrophy, increased senescence, cardiac fibroblast and endothelial dysfunction and decreased cardiac function. In particular, senescent cardiomyocytes show augmented cellular stressors, such as inflammation, ROS levels, DNA damage, endoplasmic reticulum stress, mitochondrial dysfunction, telomere shortening and SASP. All these factors determine contractile dysfunction loss and hypertrophic growth, which negatively affect myocardial function [35,36,37,38,39]. Of note, it was observed that patients who develop ventricular arrhythmias after acute myocardial infarction exhibit cardiomyocytes with a senescent phenotype characterised by increased telomere shortening [40]. Moreover, human appendages from patients with atrial fibrillation (AF) and in sinus rhythm show a higher expression of senescence markers such as p53 and p16 compared to controls, which correlate with prothrombotic and inflammatory protein expression, providing evidence of a strong correlation between AF progression and human atrial senescence [41].

In addition, an ageing heart may undergo fibrotic remodelling with senescent fibroblasts showing changes in the expression of genes regulating inflammation, extracellular matrix organization and angiogenesis. This leads to a higher proinflammatory response, autophagy and contractile dysfunction [42,43].

Senescence could also alter myofibroblast functionality: several studies have shown that senescent myofibroblasts, positive to senescence-associated beta-galactosidase (SA-β-gal), accumulate in the perivascular fibrotic areas of transverse aortic constriction-treated mice [44,45] and in the heart after myocardial infarction [45]. This was accompanied by the upregulation of the senescence regulator, p53, which was significantly upregulated in the infarcted heart or hypoxia-treated fibroblasts [45].

The dysregulation of p53 also appears to be involved in other age-related pathologies, such as hypertrophy cardiopathy. Specifically, it was found that p53 upregulation in pathological hypertrophy leads to the ubiquitylation and proteasomal degradation of Hypoxia-inducible factor 1α (HIF1α) through the E3 ubiquitin-protein ligase, MDM2, promoting heart failure [46,47,48].

Several pieces of evidence have demonstrated that senescence plays a key role in promoting vascular EC dysfunction through the dysregulation of the cell cycle, oxidative stress, altered calcium signalling and vascular inflammation (Figure 1). Senescent EC usually shows a flatter and enlarged phenotype with a polypoid nucleus. These changes could affect cytoskeleton structure, angiogenesis, proliferation and cell migration [18,20,49,50,51]. Of note, senescent EC shows increased endothelin-1 (ET-1), which promotes inflammation, the impairment of vascular relaxation and collagen accumulation [34,52,53,54]. At the same time inflammation, oxidative stress and reduced NO availability represent the main drivers of EC senescence, as reported in a recent study showing that prolonged Tumour Necrosis Factor α (TNF-α) exposure induces EC premature senescence, which is then blunted by the inhibition of NF-κB activation [20,55,56,57].

Diabetes, which is strictly associated with inflammation, oxidative stress and impaired endothelial function, has also been shown to accelerate vascular senescence and, consequently, favour the occurrence of CVD [19,20,21,22,58].

Several in vitro studies demonstrated that high-glucose-induced senescence in EC is accompanied by a reduction in NO synthesis [59,60], and interestingly, the upregulated expression of p22phox, a nicotinamide adenine dinucleotide phosphate (NADPH) oxidase component, which increases superoxide production, seems to be involved [61]. Moreover, high blood glucose levels also promote the accumulation of advanced glycation end products (AGEs) and, consequently, an increase in glycated proteins [62]. The latter has been associated with premature EC senescence and a reduction in NO bioavailability despite a three-fold increase in endothelial nitric oxide synthase (eNOS) expression [63].

Of note, animal models, such as Zucker diabetic rats and streptozotocin diabetic mice, show vascular cell senescence associated with an increase in SA β-Gal activity and p16INK4a and p53 expression [64,65,66,67,68,69,70].

In addition, several in vitro studies have highlighted the key role of the deacetylase Sirtuin 1 (SIRT1) in p53 activation through the regulation of acetylation. Indeed, SIRT1 leads to the inactivation of p53-driven DNA damage responses, which plays a major role in contributing to endothelial senescence [19,71,72,73,74]. Notably, the link between SIRT1 and p53 appears to be crucial in the phenomenon known as endothelial glycaemic memory, which, as better described in the following paragraph, contributes to the occurrence of CVD in diabetic patients, even when glycaemic control has been achieved [75,76].

### Glycaemic Memory and Endothelial Senescence

The findings of the Diabetes Control and Complications Trial (DCCT), the follow-up observational Epidemiology of Diabetes Interventions and Complications (EDIC) and 10-year UKPDS follow-up studies in patients with type 1 (T1D) or type 2 (T2D) diabetes mellitus suggest that early exposure to hyperglycaemia prompts individuals to the development of diabetic complications—a phenomenon referred to as glycaemic memory [75,76,77]. According to that, EC exposed to a hyperglycaemic environment keep showing an altered phenotype even when an ideal glycaemic control has been achieved, contributing to the development of vascular complications in diabetic patients [6,78,79,80,81].

Notably, post-translational modification, such as the acetylation of p53, has been demonstrated to be involved in the “glycaemic memory” correlated with endothelial senescence. In detail, a study conducted by Zhang and colleagues showed that transient hyperglycaemic stress induces a persistent premature cellular senescence in EC, despite the subsequent restoration of normoglycaemia. This phenomenon was mediated by the deacetylase, SIRT1, and the acetyltransferase, p300, which modulate p53 activity. Specifically, three days of high-glucose incubation followed by three days of normal glucose induced a persistent downregulation of SIRT1 and the upregulation of acetyltransferase p300, which in turn drives the hyperacetylation and activation of p53, contributing to the maintenance of p53/p21-mediated senescent “memory” [73].

In this regard, our recent study contributed to strengthening the concept of the glycaemic memory of endothelial senescence. Thoroughly, we observed that EC isolated from the umbilical cords of gestational diabetic (GD-HUVECs) women compared to EC isolated from control women revealed an impaired antioxidant enzymatic defence, as well as increased basal and stimulated ROS levels associated with decreased nuclear localization of nuclear factor erythroid 2–related factor 2 (Nrf2)—an essential transcriptional factor regulating the antioxidant defence gene expression [82]. Furthermore, we found that GD-HUVEC, exposed to in vivo chronic hyperglycaemia, shows reduced SIRT1 activity, together with an increase in p300-mediated p53 acetylation compared to control cells. Finally, in agreement with the data from Zhang and colleagues, we demonstrated that p300 silencing reduced both the high-glucose-increased protein levels of p300 and acetylated p53 in control cells and their elevated basal levels in GD-cells, thus, indicating the possible involvement of SIRT1/P300/P53/P21 pathway in the early senescent GD-HUVEC phenotype [80].

Furthermore, several studies highlighted the role of p300 in establishing glycaemic memory also through epigenetic modifications, such as histone acetylation [79,83,84,85,86]. In the DCCT and EDIC trials, chromatin and DNA analysis performed in T1D patients revealed that monocytes from patients who developed complications during the subsequent EDIC follow-up study showed significant enrichment of histone 3 lysine 9 acetylation (H3K9ac), a gene-associated activation mark, located at key inflammatory loci [87,88]. In support of this evidence, it has been observed that prolonged exposure to specific inflammatory stimuli, such as TNF-α or lipopolysaccharides, determines an increase in histone 3 lysine 27 acetylation (H3K27ac) mediated by p300 at the gene enhancers imprinting a memory, which induce these cells to respond faster and stronger to a second inflammatory stimulation [89].

Therefore, we believe that a deep discernment of p300 function in inflammation and oxidative stress, the two main drivers of diabetes-related endothelial dysfunction, is urgently needed to support new pre-clinical and clinical studies.

## 3. Protein Lysine Acetylation by p300 in Diabetes

Several studies have pointed out the key role of acetylation in diabetes [90], which is a reversible reaction consisting of the transfer of an acetyl group from acetyl-coenzyme A (acetyl-CoA) to the ε-amino group of lysine residues [91,92,93]. Histone lysine acetylation is an epigenetic modification that is able to regulate gene transcriptional activity by inducing chromatin relaxation [91]. However, lysine acetylation also occurs in non-histone proteins, which modulate cellular processes, including gene transcription, cell cycle, cell division, DNA damage repair, signalling transduction, protein folding, protein aggregation and autophagy [94,95,96].

Acetylation is mainly controlled by histone acetyltransferases (HATs) and histone deacetylases (HDACs). HATs are classified into the following families: general control non-depressible 5 (GCN5), CREB-binding protein (CBP)/p300 and Moz, Ybf2/Sas3, Sas2 and Tip60 (MYST families). While the NAD^+^-dependent SIRT family is the most common type of HDAC [97].

The acetyltransferase, p300, regulates several cellular processes, including proliferation, migration, differentiation, senescence and apoptosis [98,99,100]. It drives histone acetylation, playing an important role as an epigenetic regulator through chromatin remodelling. In detail, p300 transfers the acetyl group from acetyl-CoA to histone lysine residues, leading to chromatin relaxation, which makes DNA more accessible for gene transcription [101,102]. It also functions as acetyltransferase for non-histone targets, such as transcription factors, enhancing their DNA-binding activity [73,80,103,104].

Importantly, p300 is involved in regulating numerous transcription factors, including NF-κB [105]. The N- and C-terminal domains of both CBP and p300 functionally interact with the region of p65 containing the transcriptional activation domain, playing an important role in the cytokine-induced expression of various immune and inflammatory genes and ECM proteins [106,107,108]. p300 also regulates myocyte enhancing factor 2 (MEF2) and GATA binding protein 4 (GATA4) [109]. Furthermore, p300 may control cellular processes by regulating protein–protein interactions as a coactivator. It is the case of the hypoxia gene transcription that requires the binding of p300 to hypoxia-inducible-factor-1alpha (HIF-1alpha) and some inflammatory gene promoters, which are bound by the p65 interacting with p300 [110,111,112].

## 4. Role of p300 in Diabetic Cardiovascular Complications

Numerous studies have documented the physiological role of p300 in development and heart diseases, such as accelerated cardiac hypertrophy, cardiomyopathy, matrix remodelling or fibrogenesis and heart failure [113]. The involvement of p300 acetyltransferase activity in mammalian heart development was highlighted by studies showing that mice with a mutation of p300, causing the lack of acetyltransferase domain, are characterised by embryonic lethality [114] and cardiomyopathy-associated cardiac dysfunction [115].

Recently, lysine acetylation has emerged as a major mechanism underlying diabetic vascular complications [116,117,118,119,120]. Indeed, supporting this idea, it has been demonstrated that acetyltransferase p300 may accelerate cardiac ageing by affecting ECM remodelling. This could be explained by the fact that p300 plays an essential role in profibrogenic cytokine transforming growth factor beta (TGF-β)-induced synthesis of type I collagen. In particular, the interaction of p300 with phospho-Smad2/3 turned out to be essential for matrix protein collagen synthesis and secretion by fibroblasts [121].

Additionally, p300 was significantly elevated in myofibroblast-like cells derived from the primary culture of mouse cardiac EC in response to profibrogenic cytokine TGF-β. On the contrary, the dissociation of p300 from the transcription initiation complex at the collagen gene promoter blunts profibrogenic signalling-induced type I collagen synthesis [122]. Interestingly, small acetyltransferase inhibitors suppress p300 activity in fibroblasts, reducing TGF-β-associated H3K9ac, myofibroblast differentiation and matrix protein synthesis [123]. Furthermore, p300 is known to control the transcription of a variety of genes involved in cardiac hypertrophy.

Gusterson and colleagues demonstrated that inhibition of p300 activity arrests cardiomyocyte hypertrophy induced by phenylephrine. On the other hand, p300 overexpression promotes this adverse cardiac phenotype [124,125]. It was also shown that phenylephrine induces the expression of p300, the acetylation of GATA4 and its binding to the ET-1 promoter in cardiomyocytes [126].

Notably, p300 levels are significantly elevated also in angiotensin (Ang) II-induced hypertensive myocardial tissues and are associated with increased H3K9ac and cardiac hypertrophy and fibrogenesis [127]. Moreover, p300-driven H3K9ac also appears to be involved in cardiomyopathy associated with loss-of-function mutations of Short-chain enoyl-CoA hydratase (ECHS1)—a key mitochondrial enzyme for fatty acid β-oxidation [128].

In vascular smooth muscle cells (VSMCs), it was also observed that p300 modulates the pro-atherogenic effect of 12(S)-Hydroxyeicosatetraenoic acid (12(S)-HETE) by mediating IL-6 and MCP-1 expression through p300-driven H3K9/14ac [129].

Recently, several studies pointed out the importance of p300 as a regulator of diabetes-associated CVD [36,85,113,130,131,132,133]. In fact, the expression of p300 was found to be remarkably elevated in the hearts of diabetic rats compared with normal controls. Accordingly, an increase in hypertrophy in neonatal rat cardiomyocytes exposed to high levels of glucose was observed [134]. Among the factors contributing to diabetic cardiopathy, there is also the accumulation of ECM that is responsible for the increase in fibrosis, in which p300 plays a key role by activating TGF-β via the acetylation of Smad2 [135]. Diabetes favours the production of ECM proteins also through the activation of NF-κB and the activator protein 1 (AP-1) signalling pathway, which appears to be regulated by p300 in diabetic rats [136]. This was also supported by an in vitro study demonstrating that p300 regulates the expression of vasoactive factors and ECM proteins in EC exposed to high glucose through histone acetylation [137].

As mentioned above, the tight interconnection between p300 and SIRT1 may regulate accelerated vascular ageing associated with hyperglycaemia. In this regard, EC exposed to high glucose and tissues from diabetic animals showed downregulation of SIRT1, which, in turn, leads to the reduction in mitochondrial antioxidant enzymes in a p300- and Forkhead box O1 (FOXO1)-mediated pathway [138].

The essential role of p300 and SIRT1 in high-glucose-induced vascular senescence was further confirmed in a cellular model of hyperglycaemia, evidencing the direct role of p300 in the activation of p53 and, consequently, p21 [73,80]. Additionally, p300 seems also to be involved in vascular tone regulation, as suggested by the fact that p300-mediated hyperacetylation of lysine residues in VSMCs has been associated with impairing VSMCs-dependent vasodilation in advanced T2D [130].

### 4.1. Inflammation

Diabetes is associated with augmented inflammation, which has been proven by a consistent body of evidence as one of the main drivers of atherosclerotic CVD [4,139] and is characterised by the increased activity of inflammasomes and increased levels of proinflammatory cytokines [140,141].

The role of p300 appears to be crucial in the regulation of NF-κB—one of the main mediators of the inflammatory pathway [142]. Lan and colleagues demonstrated in a T2D nephropathy mouse model that p300 leads to the activation of the NF-κB signalling pathway and the production of proinflammatory cytokines by a direct interaction with the subunit, p65 [103,143]. In diabetic mice, it was also shown that p300-driven H3K9/14ac modulates c-Jun N-terminal kinase (JNK)-downstream genes, such as connective tissue growth factor (CTGF), plasminogen activator inhibitor-1 (PAI-1) and fibronectin 1, and that the inhibition of p300 by a curcumin analogue (C66) was able to prevent renal injury and dysfunction in diabetic mice [144].

Several in vitro studies supported the key role of p300 in vascular inflammation associated with hyperglycaemia. Chen and colleagues demonstrated that glucose exposure increased p300 expression, histone acetylation and p300 binding to ET-1 and to fibronectin promoters in EC. This suggests that glucose-induced p300 upregulation regulates the gene expression of vasoactive factors and ECM proteins in EC through epigenetic mechanisms [137].

Moreover, in vitro studies have shown that the p300-driven acetylation of H3K9 recruits NF-κB to the promoters of proinflammatory genes, such as IL-6, IL-8 and cyclooxygenase-2 (COX-2) [145,146].

It was also found that high glucose increases the p300-driven acetylation of histones H3K14, H3K18, H3K23, H4K5 and H4K16 in retinal glial cells. These changes are positively correlated with the induction of the proinflammatory molecules ICAM-1, Inducible Nitric Oxide Synthase (iNOS) and Vascular Endothelial Growth Factor (VEGF). Of note, both the acetylation and expression of the inflammatory proteins can be inhibited by modifying HAT/HDAC activity [117].

The regulating Forkhead box O3a (FOXO3a) acetylation by p300 is another mechanism involved in diabetes-induced inflammation: diabetes-induced FOXO3a acetylation prevents its binding to gene target promoters, increasing NOD-, LRR- and pyrin domain-containing protein 3 (NLRP3) inflammasome activation. This mechanism was then reverted by C646, a p300 inhibitor [147].

### 4.2. Oxidative Stress

Diabetes enhances oxidative stress and compromises antioxidant defences, as evidenced by increased superoxide generation and reduced Superoxide dismutase (SOD) activity catalase (CAT) and glutathione (GLT) [148]. Chronic hyperglycaemia also diminishes endothelial NO bioavailability, resulting in vascular homeostasis impairment, which, in turn, leads to a prothrombotic and proinflammatory state [149].

It is well known that eNOS plays a central role in vascular homeostasis by regulating the synthesis of NO and that the impairment of enzyme activity is involved in the pathogenesis of diabetic endothelial damage [150]. In this regard, histone acetylation appears to be highly implicated in the regulation of eNOS gene expression in EC; H3K9ac and H4K12ac at the eNOS promoter are functionally relevant to its expression, as confirmed by the treatment of cells with trichostatin A, a histone deacetylase inhibitor, which was associated with the increased acetylation of histones H3 and H4 at the eNOS proximal promoter [151].

In support of these results, the involvement of p300 in the regulation of eNOS was demonstrated. In detail, Chen and colleagues have shown that the acute stimulation of endothelial eNOS mRNA transcription by laminar shear stress is dependent on the NF-kB subunits, p50 and p65, which in turn bind to a shear stress response element (SSRE) in the human eNOS promoter. This is mediated by the activation of p300, which acetylates p65 and the histones, H3 and H4, in proximity to the human eNOS SSRE, promoting an opened chromatin structure and allowing gene transcription [152].

A recent study indicated a key role of histone acetyltransferase p300 in diabetes-accelerated renal damage. Interestingly, the authors observed that kidneys of diabetic C57BL/6J mice treated with C646, a pharmacological inhibitor of p300, showed significant downregulation of H3K27ac and diabetes-induced expression of NADPH oxidase (NOX), known to be the main trigger of ROS production [153].

SIRT1, antagonizing p300, has been shown to inhibit transcriptional factors, such as NF-κB, MAPK, Matrix metallopeptidase 9(MMP-9), FOXO3a and p53. On the other hand, it increases Cardiac sarcoplasmic reticulum Ca2+-ATPase2a (SERCA2a), extracellular signal-regulated protein kinase (ERK1/2), eNOS activity, proliferator-activated receptor-gamma coactivator α (PGC-1α) and AMP-activated protein Kinase (AMPK) [154]. The overexpression of SIRT1 was also able to decrease superoxide production and increase antioxidant enzyme SOD activity in diabetic ischaemic reperfused heart tissue [155]. Importantly, as mentioned above, SIRT1 plays a major role also in the regulation of eNOS activity by reducing eNOS acetylation and increasing its activation [156].

In VSMCs, HDAC1/2 and p300 proteins were found to bind the promoter’s sites of active NOX1/4/5 genes. H3K27 acetylation was increased at the promoters of NOX genes in high-glucose-exposed VSMCs [157]. 

## 5. Histone Acetyltransferase Inhibitors: Future Prospective

Considering the possibility of interfering with epigenetic modifications [131], Histone acetyltransferase inhibitors (HATis) could be explored as new potential drugs targeting the epigenetic mechanisms involved in vascular damage associated with diabetes [158].

Some compounds, such as metformin, already used in clinical practice and established as a first-line drug in the treatment of T2D patients, have shown some epigenetic effects [159]. Metformin may inhibit HATs through the activation of AMPK [160]. Furthermore, a recent study demonstrates that metformin can restrain phenylephrine-induced hypertrophic responses by inhibiting p300 activity in cardiomyocytes [161].

Interestingly, the activity of some pharmacological inhibitors targeting epigenetic modifiers for the treatment of cancer, such as JQ1 [162], has also been investigated in experimental models of diabetes complications. Notably, it was able to inhibit An-gII-regulated genes in vitro, as well as AngII-induced hypertension in mice [86].

To date, studies on the potential effect of drugs with proven cardio-renal benefits in diabetes, such as glucagon-like peptide 1 agonists (GLP1-as) and sodium-glucose co-transporter-2 inhibitors (SGLT2is), on p300 regulation are lacking. However, a recent study showed that liraglutide, a GLP1-a that has been reported to exert protective effects against myocardial ischaemia–reperfusion injury, displays a beneficial effect on renal ischaemia–reperfusion injury by decreasing histone acetyltransferase activity [163].

Regarding SGLT2i, the new drugs, β-hydroxybutyrate and dapagliflozin, have shown some effects on oxidative stress and cardioprotection, respectively, through epigenetic mechanisms [164,165]. Therefore, this evidence supports a further investigation to better define the effect of these drugs on the regulation of p300 expression and/or activity.

Among natural compounds, there is garcinol—a potent p300 inhibitor—that was able to mitigate high-glucose-induced inflammation in retinal cells, suggesting its potential use in the prevention of diabetic retinopathy [117,166].

A recent study demonstrated that epigallocatechin gallate (EGCG), the major bioactive polyphenol present in green tea, attenuates vascular inflammation via a repressive epigenetic effect on the NF-κB signalling pathway. In detail, this compound was able to block the recruitment of p65, p300 and HDAC1-3 in the promoters of IL-6, C-reactive Protein (CRP), ICAM-1, vascular cell adhesion molecule 1 (VCAM-1), IL-1α and IL-1β [142].

Curcumin has been considered a potential epidrug for diabetes because of its hypoglycaemic, hypolipidaemic and epigenetic effects in various rodent models [167]. Interestingly, a study conducted by Jung-Mi Yun and colleagues suggests that curcumin could improve glucose metabolism and prevent diabetic complications by activating HDAC-2 and inhibiting p300, which, in turn, reduce NF-κB signalling and vascular inflammation [168]. Furthermore, curcumin, through the modulation of the PKC-α and MAPK pathways, decreased the mRNA expression of transcriptional coactivator p300, the accumulation of ECM proteins and controlled oxidative stress and apoptosis in the heart of diabetic rats; this resulted in the attenuation of cardiomyocyte hypertrophy, myocardial fibrosis and left ventricular dysfunction [169].

Unfortunately, important evidence gaps remain before implementing the use of these substances in a clinical setting. These are mainly due to the lack of specificity in chromatin modifiers, implicating the unspecific and undesirable modulation of transcriptional programs with wide effects on cellular pathways. In addition, the safety of these approaches remains to be proven.

## 6. Conclusions

In summary, the evidence analysed and reported in this review supports the idea that p300 may be a relevant actor in diabetes-induced vascular inflammation and oxidative stress, which promote endothelial senescence, driving atherosclerotic cardiovascular disease.

As shown in Figure 2, it may modulate vascular alterations through both epigenetic mechanisms and transcription factor acetylation. Specifically, p300 may regulate the activation of the inflammatory signalling pathway by interacting directly with NF-κB or by inducing its acetylation. This suggests a crucial role of p300 as an important bridge between NF-κB p65 and the transcriptional machinery. Recently, the role of p300 as an epigenetic modifier that is able to regulate the upstream activation of inflammatory cytokines and oxidative stress in the setting of CV complication in diabetes has also emerged. Indeed, recent evidence unveiled a new function of p300 driving histone acetylation in establishing the so-called glycaemic memory, which turned out to be a key process in establishing permanent vascular damage.

Lastly, several in vitro and in vivo pre-clinical studies shed light on the potential use of acetyltransferase inhibitors, which, following further clinical validation, could be promising in the treatment of diabetes complications. Therefore, it would be very interesting to further study p300 as a potential therapeutic target that may contribute to the clinical management of diabetic vascular complications.

## Figures and Tables

**Figure 1 cells-12-00431-f001:**
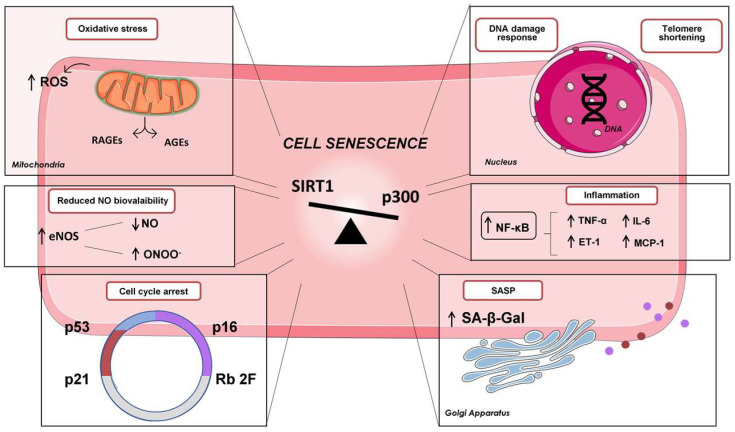
Endothelial cell senescence: Senescence is triggered by various stimuli, including changes in telomeric structure, oxidative stress, mitochondrial dysfunction, inflammation and DNA damage, which contribute to cell cycle arrest through the activation of the p53/p21CIP1 or p16INK4A/hosphor-Rb tumour suppressor pathways. In addition, senescent cells exhibit activation of the senescence-associated secretory phenotype (SASP) characterised by the increased lysosomal activity and accumulation of β-galactosidase. The SASP leads to the upregulation and release of growth factors, cytokines and proteases that can exert detrimental effects.

**Figure 2 cells-12-00431-f002:**
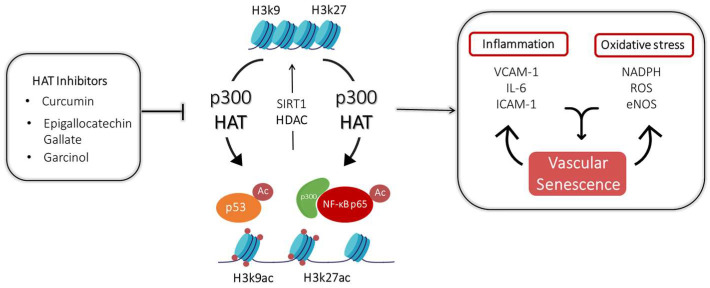
p300 mechanisms underlying vascular inflammation and oxidative stress and the role of some HAT inhibitors.

## Data Availability

Not applicable.

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
