# Peer review of "Diabetes and Its Cardiovascular Complications: Potential Role of the Acetyltransferase p300"

_cells, 2023, doi:10.3390/cells12030431_

Round 1

Reviewer 1 Report

I read with great interest the paper “Diabetes and its cardiovascular complications: potential role of the acetyltransferase p300” by Nadia Di Pietrantonio et al.

The design is fine. The article is logically divided into sections and subsections. It is an interesting review on the potential role of acetyltransferase p300 in cardiovascular complications development in diabetes. Figures are nice.

Comments:

1.       The role of p53 has been deeply reviewed throughout the paper. However, there is another important issue that should be treated. In fact, the upregulation of p53 is also involved in the ubiquitylation and proteasomal degradation of Hypoxia-inducible factor 1α (one of the major transcription factors that regulates oxygen homeostasis through angiogenesis, vascular remodelling, and glucose metabolism), probably through the E3 ubiquitin-protein ligase MDM2, which in turn leads to an unbalance between myocardial growth and capillary density, promoting cardiovascular disease onset and progression (doi: 10.31083/J.RCM2305165).

2.       Cardiovascular disease and diabetes: the pathogenesis of endothelial dysfunction is poorly reported and there are some flows. Please revise.

3.       Histone acetyltransferase inhibitors: future prospective: in this paragraph the authors have provided evidence about metformin, an old drug that has several pleiotropic effects and has also proven to ameliorate endothelial dysfunction. Is there any evidence about gliflozins and/or GLP1a in p300 expression?

Author Response

Reviewer 1

 I read with great interest the paper “Diabetes and its cardiovascular complications: potential role of the acetyltransferase p300” by Nadia Di Pietrantonio et al.

The design is fine. The article is logically divided into sections and subsections. It is an interesting review on the potential role of acetyltransferase p300 in cardiovascular complications development in diabetes. Figures are nice.

We thank Reviewer #1 for the appreciation of our manuscript as well as for his/her useful comments in response to which changes have been made. Particularly, the sections modified or added in the text are highlighted in red as track changes mode.

Comments

1.The role of p53 has been deeply reviewed throughout the paper. However, there is another important issue that should be treated. In fact, the upregulation of p53 is also involved in the ubiquitylation and proteasomal degradation of Hypoxia-inducible factor 1α (one of the major transcription factors that regulates oxygen homeostasis through angiogenesis, vascular remodelling, and glucose metabolism), probably through the E3 ubiquitin-protein ligase MDM2, which in turn leads to an unbalance between myocardial growth and capillary density, promoting cardiovascular disease onset and progression (doi: 10.31083/J.RCM2305165).

We are grateful to the Reviewer for giving us the opportunity to better clarify an important aspect of p53 function. Therefore, in the revised version of the paragraph “Cardiovascular senescence” we have added a part addressing the issue of the involvement of p53 in the regulation of Hypoxia-inducible factor 1α, highly involved in vascular remodelling (lines 127-131).

  1. cardiovascular disease and diabetes: the pathogenesis of endothelial dysfunction is poorly reported and there are some flows. Please revise.

The Reviewer suggestion gives us the opportunity to better clarify the pathogenesis of endothelial dysfunction, a key initiating process of diabetes associated cardiovascular complication. Therefore, we described more in detail this process by highlighting the main alterations which impair the normal functionality of the endothelium in the paragraph “Cardiovascular disease and diabetes” (lines 42-64).

  1. Histone acetyltransferase inhibitors: future prospective: in this paragraph the authors have provided evidence about metformin, an old drug that has several pleiotropic effects and has also proven to ameliorate endothelial dysfunction. Is there any evidence about gliflozins and/or GLP1a in p300 expression?

We are grateful for your comment, which arises an important issue unfortunately still barely addressed in the literature. In fact, studies on the potential effect of GLP1-RAs and SGLT2i on p300 expression/activity are lacking. However, a recent study showed that liraglutide, a glucagon-like peptide-1 receptor (GLP-1R) agonist which has been reported to exert protective effects against myocardial ischemia-reperfusion injury, displays a beneficial effect on renal ischemia-reperfusion injury by decreasing histone acetyltransferase activity and acetylation of High Mobility Group Box 1 protein (HMGB1) in mice (PMID: 34481074 DOI: 10.1016/j.phrs.2021.105867).

Although, no evidence on the effect of SGLT2i on p300 have been reported, there are some interesting findings unveiling new potential mechanism of these drugs. In particular, β-hydroxybutyrate has been reported to inhibits class I HDACs through histone acetylation and increases oxidative stress resistance (PMCID: PMC3735349 DOI: 10.1126/science.1227166). Additionally, it was recently found that the SGLT2 inhibitor dapaglifozin exerts a cardio protective effect through an epigenetic mechanism (PMID: 31162549 DOI: 10.1210/jc.2019-00706). The role of these new drugs, with proven cardio-renal benefit, in the regulation of p300 is very interesting and needs to be further investigated in future studies.

In this regard we added a sentence in the paragraph “Histone acetyltransferase inhibitors: future prospective” (lines 394-404).

Reviewer 2 Report

Diabetes is a major risk factor of macro-complications. Clinical studies show that diabetic patients have twice of chance to develop cardiac complication than the healthy control population. In addition, more than 2/3 of diabetic patients develop cardiovascular disease and may die from it. Therefore, diabetes related cardiovascular complications are of paramount clinical interest, which is also an unmet problem.

In this review manuscript, the authors discussed diabetic complications from an interesting angle. The authors aimed to link epigenetic modifications, inflammation, and diabetic complications. Although both the topic and theme are quite interesting, a number of flaws preclude its acceptance to the field.

1. Throughout the review, the authors tried to tie senescence to diabetic cardiovascular complications. However, my reading of the abstract indicates otherwise. Senescence was mentioned only once in the abstract. Therefore, there seems a disconnection between the abstract and the body of the review.

2. While I admit that senescence is an important contributor of diabetic cardiovascular complications, it may not be the most essential one. In section 1, I would hope to see more broad outline of diabetic cardiovascular complications and underlying pathologies. The author can narrow down and focus on senescence in section 2.

3. Senescence itself is an active research topic not only in diabetes but also in other fields, such as cancer. It would be preferred if the authors could summarize senescence in a more broad point of view in depth. The current description in section 2 paragraph 1 seems short of details.

4. In addition to endothelial cell senescence, senescence also happens in other cell types, such as myofibroblasts in the conditions of myocardial infarction, aortic constriction, ischemia/reperfusion, etc. These aspects should be discussed.

5. There are a number of grammatical mistakes. An carefully proofreading is recommended.

Author Response

Reviewer 2

Diabetes is a major risk factor of macro-complications. Clinical studies show that diabetic patients have twice of chance to develop cardiac complication than the healthy control population. In addition, more than 2/3 of diabetic patients develop cardiovascular disease and may die from it. Therefore, diabetes related cardiovascular complications are of paramount clinical interest, which is also an unmet problem.

In this review manuscript, the authors discussed diabetic complications from an interesting angle. The authors aimed to link epigenetic modifications, inflammation, and diabetic complications. Although both the topic and theme are quite interesting, a number of flaws preclude its acceptance to the field.

We thank Reviewer #2 for the appreciation of our manuscript as well as for his/her useful comments in response to which changes have been made. Particularly, the sections modified or added in the text are highlighted in red as track changes mode.

Comments

  1. Throughout the review, the authors tried to tie senescence to diabetic cardiovascular complications. However, my reading of the abstract indicates otherwise. Senescence was mentioned only once in the abstract. Therefore, there seems a disconnection between the abstract and the body of the review.

We thank the Reviewer for this observation. Therefore, in the revised manuscript we modify the abstract to better clarify the main topic of our review.

2.While I admit that senescence is an important contributor of diabetic cardiovascular complications, it may not be the most essential one. In section 1, I would hope to see more broad outline of diabetic cardiovascular complications and underlying pathologies. The author can narrow down and focus on senescence in section.

As suggested by the Reviewer, we added in the paragraph “Cardiovascular disease and diabetes” a section describing the main cardiovascular complications associated to diabetes (lines 34-38). Then we better described all the factors which may affect endothelial dysfunction in the setting of diabetes, including hypeglycemia. All these factors (hyperglycemia, free fatty acids and triglycerides) may alter different aspects of the endothelium homeostasis though different mechanisms that have been detailed in the section describing endothelial dysfunction pathogenesis (lines 42-64).

Additionally, it is demonstrated that diabetic patients present dysfunction of multiple organ systems similarly to that observed in chronological aging, suggesting that this metabolic disease might induce an early aging state associated with premature vascular senescence, implicated in the development of endothelial dysfunction (lines 71-73).

  1. Senescence itself is an active research topic not only in diabetes but also in other fields, such as cancer. It would be preferred if the authors could summarize senescence in a more broad point of view in depth. The current description in section 2 paragraph 1 seems short of details.

In order to address this comment, we added a section in the paragraph “Cardiovascular senescence and diabetes” (lines 89-97), providing a broader overview of the degenerative pathologies which may be driven by age related alterations.

4.In addition to endothelial cell senescence, senescence also happens in other cell types, such as myofibroblasts in the conditions of myocardial infarction, aortic constriction, ischemia/reperfusion, etc. These aspects should be discussed.

We appreciated the Reviewer for pointing out the lack of an overview on the role of senescence in other cell types. To address this comment, we added a section in the paragraph “Cardiovascular senescence and diabetes” (lines 103-126) which highlights the involvement of senescence mechanisms in the dysfunction of cell types such cardiomyocytes, fibroblasts and myofibroblast.

5.There are a number of grammatical mistakes. An carefully proofreading is recommended

English and grammatical mistakes have been revised carefully in all the manuscript and can be highlighted in red as track changes mode.

Round 2

Reviewer 2 Report

The authors have addressed my previous questions. I do not have additional concerns.